# Nutritional Challenges in Pregnant Women with Renal Diseases: Relevance to Fetal Outcomes

**DOI:** 10.3390/nu12030873

**Published:** 2020-03-24

**Authors:** Pasquale Esposito, Giacomo Garibotto, Daniela Picciotto, Francesca Costigliolo, Francesca Viazzi, Novella Evelina Conti

**Affiliations:** Clinica Nefrologica, Dialisi, Trapianto, Department of Internal Medicine, University of Genoa and IRCCS Ospedale Policlinico San Martino, Viale Benedetto XV, 16132 Genoa, Italy; gari@unige.it (G.G.); danipicciotto@me.com (D.P.); 3232594@studenti.unige.it (F.C.); francesca.viazzi@unige.it (F.V.); novella.conti@hsanmartino.it (N.E.C.)

**Keywords:** pregnancy, nutrition, chronic kidney disease, fetal outcomes, dialysis, kidney transplantation

## Abstract

Pregnancy in women affected by chronic kidney disease (CKD) has become more common in recent years, probably as a consequence of increased CKD prevalence and improvements in the care provided to these patients. Management of this condition requires careful attention since many clinical aspects have to be taken into consideration, including the reciprocal influence of the renal disease and pregnancy, the need for adjustment of the medical treatments and the high risk of maternal and obstetric complications. Nutrition assessment and management is a crucial step in this process, since nutritional status may affect both maternal and fetal health, with potential effects also on the future development of adult diseases in the offspring. Nevertheless, few data are available on the nutritional management of pregnant women with CKD and the main clinical indications are based on small case series or are extrapolated from the general recommendations for non-pregnant CKD patients. In this review, we discuss the main issues regarding the nutritional management of pregnant women with renal diseases, including CKD patients on conservative treatment, patients on dialysis and kidney transplant patients, focusing on their relevance on fetal outcomes and considering the peculiarities of this population and the approaches that could be implemented into clinical practice.

## 1. Pregnancy in Women with Renal Disease: General Considerations

The prevalence of chronic kidney disease (CKD) in the general population is increased in recent years, and now it has been calculated that 3–4% of women of childbearing age are affected by CKD [1]. Paralleling, the pregnancy rate in CKD women has raised over time, also as a consequence of better CKD management and improvement in the antenatal care provided to these high-risk pregnancies [2]. Available data from large studies show that women with CKD, compared with women without CKD, have worse maternal and neonatal outcomes, mainly because pregnancy-related physiologic, hemodynamic and metabolic changes are less efficient in CKD patients [3]. Maternal and fetal complications include gestational hypertension, polyhydramnios, intrauterine growth restriction, and superimposed pre-eclampsia that is associated with small for gestational age (SGA) babies and preterm birth [4]. Interestingly, since fetal renal maturation may be stopped in the case of early delivery, paradoxically, babies of women with CKD may present an increased risk to develop renal diseases later in life [5]. 

Several pieces of research have been performed to define the risk factors associated with adverse pregnancy outcomes in CKD. In the Torino-Cagliari Observational Study (TOCOS), a large prospective study, pregnancy outcomes were evaluated in 504 pregnancies with CKD vs. 836 low-risk pregnancies in women without CKD. The authors found that preterm delivery was associated with CKD stage, baseline hypertension, the presence of systemic disease and proteinuria. Interestingly, when compared with control subjects, the pregnancy outcome results were also worse in patients with stage 1 CKD (i.e., with still normal renal function), even in the absence of other comorbidities, thus suggesting that CKD per se might represent a risk factor for adverse pregnancy-related outcomes [6]. Moreover, looking at pregnancy in CKD, it should be also considered that this condition can constitute a trigger for the progression of maternal renal disease.

A prospective study on 49 pregnancies in CKD patients (with basal GFR <40 mL/min and proteinuria >1 g per day) showed an acceleration in renal disease progression after pregnancy, with a mean GFR reduction of 1.17 mL/min per month, compared with a pre-pregnancy reduction of about 0.20 mL/min per month [7]. However, a global risk assessment of CKD pregnant patients is further complicated by the fact that CKD may be caused by many different renal diseases that could have specific clinical features. These diseases include genetic disorders, primary and secondary glomerulonephritis, diabetes kidney disease, etc. [8] Only a few studies have explored the risk assessment of maternal and pregnancy outcomes in a specific renal disease. For example, it has been proved that in pregnant women with immunoglobulin A (IgA) nephropathy proteinuria and birthweight are negatively correlated, while proteinuria >1 g per day is associated with the loss of renal function [9].

In this complex setting, pre-pregnancy counseling is necessary and useful to inform women about the potential maternal and fetal pregnancy-related risks [10]. Furthermore, strict and multidisciplinary follow-up is mandatory to provide the best balance between maternal and fetal needs, to identify and manage complications and plan delivery [11].

## 2. Pregnancy in Women with Renal Disease: Nutritional Issues

Good nutrition plays a substantial role in healthy pregnant patients, who need to face the many physiologic and metabolic adaptations that occur during pregnancy, avoiding both malnutrition and overnutrition [12]. In particular, maternal nutritional status plays a critical role in the fetus health [13]. Indeed, a diet containing sufficient amounts of macro- and micronutrients is essential to fetus organogenesis, whereas adverse nutritional conditions during pregnancy may permanently change the structure and function of specific organs, exposing the offspring also to the risk to develop adult diseases [14]. The difficulty to provide the correct nutrition in pregnant women is further complicated in CKD patients, in whom nutritional therapy is a cornerstone of the clinical management of renal diseases [15]. So, in pregnant CKD women, there is the need to maintain a constant balance between the specific pregnancy- and disease-related nutritional needs, aiming to achieve both optimal nutrient intakes to promote fetal growth and development, and adequate metabolic control of the renal disease. Nutritional assessment and implementation of a nutritional plan should be performed early after the diagnosis of pregnancy because the first months are crucial for fetal development. The first step is to define the nutritional needs of macronutrients, micronutrients, water, and electrolytes. In this regard, data on pregnant CKD patients are lacking and most of the indications derived from that applied from CKD nutrition treatment (without pregnancy) and pregnancy nutrition recommendations for the healthy women [16]. 

Nutritional management in CKD pregnant patients may be complex and requires a comprehensive understanding of the particular characteristics and needs of this patient population. 

In this review, we discuss the main issues about nutritional management of CKD pregnant women in different CKD scenarios: conservative management, dialysis, and kidney transplantation, highlighting the influences on fetal outcomes and the peculiarities of this population and the clinical approaches that could be implemented into clinical practice.

## 3. CKD Non-Dialysis

### 3.1. Pregnancy in CKD Non-Dialysis Women 

As stated above, CKD pregnant patients show highly heterogeneous clinical features, depending on the severity of renal disease (i.e., CKD stages) and type of kidney disease. So, immunologic diseases can flare up more easily during pregnancy and have a higher risk of developing heavy proteinuria and hypertension. This is the case of pregnant women with systemic lupus erythematosus with renal involvement, i.e., lupus nephritis, in whom an increased disease severity represents the major risk factor for prematurity [17]. Other renal diseases potentially affecting pregnancy outcomes include diabetic kidney disease, which is related to a higher risk of fetal malformation and perinatal death rates [18], and adult polycystic kidney disease that show a high prevalence of urinary infections and preterm delivery [19]. Moreover, there is a lack of a shared assessment (also due to difficulty to evaluate renal function) and management of CKD in pregnancy, mainly because performing clinical trials on pregnant patients presents obvious ethical issues [20]. 

### 3.2. Principles of Nutritional Management in Pregnant Women and Non-Dialysis Treated CKD

Nutrition plans in pregnant CKD women should take into consideration that the increased need for energy, protein, vitamin, and minerals of pregnancy must be carefully balanced with CKD features as altered serum electrolytes levels, changes in volume status, blood pressure values, and use of multiple medications could also affect fetal growth and maturation.

Regarding the global energy requirements, there is a lack of data in CKD pregnant women in stages 1 and 2 in whom it seems prudent to follow the recommendations for healthy pregnancies. Instead, in women with more advanced renal disease (i.e., CKD 3-5), who usually present a high resting energy expenditure, it has been suggested to follow the general advice used for CKD patients [21]. So, it is recommended an energy intake of about 35 Kcal/kg/day, calculated on pre-pregnancy weight [22]. To this basal amount, the specific pregnancy-related energy cost should be added, which has been estimated as 85 kcal/day, 285 kcal/day, and 475 kcal/day during the 1st, 2nd, and 3rd trimester, respectively [23]. Alternatively, a more simple and practical approach suggests adding to the basal diet about 300 kcal/day from the second trimester. This kind of approach has been related to a lower incidence of neonatal malnutrition [24]. 

However, all these suggestions should be personalized according to the individual needs, taking into account that, beyond undernutrition, also obesity (mainly occurring in high-income countries) constitutes a risk factor for poor fetal outcomes and should be avoided [25]. 

As for healthy pregnancy, in CKD patients the recommended fat intake is 30–35% of the total calories, while acceptable carbohydrate recommended intake is about 45–65% [11]. 

Management of protein intake is more difficult since it is strongly influenced by CKD stage and it may have a significant impact on the metabolic control of CKD patients. This is true mainly for advanced CKD (stages 4–5) when a low protein diet is usually recommended as a part of the conservative approach of the renal disease. 

Otherwise, the important role of protein intake in organ, and in particular, in renal development is witnessed by data from animal models in which starvation and extreme protein intake during pregnancy leads to intrauterine growth retardation, reduction of the nephron number and alteration of kidney morphology [26,27]. Similarly, clinical experiences in developing countries have shown that undernutrition and very low protein regimens have harmful effects on fetal growth [28].

Due to the potential high clinical impact, many pieces of research have focused on the study of the feasibility of low protein diets in pregnant CKD women, finding that balanced low protein/vegan diets supplemented with amino acids (detailed below) are safe in these patients [29,30]. So, current expert opinions recommend in pregnant non-dialysis dependent CKD women in stages 1 and 2 a protein intake similar to that for a healthy pregnancy, and in CKD stages 3–5, a moderate protein-restricted plant-based diet (0.6–0.8 g/kg/ day of protein) plus protein supplementation of about 10 gr/day (mainly provided by keto-analogues) [31]. 

These recommendations and the main nutritional needs of pregnant CKD women are summarized in Table 1.

### 3.3. Clinical Experiences

First of all, nutritional counseling is essential to understand the individual nutritional habits and warrant an adequate caloric, protein, and other nutrients intake. In particular, it is important to assist patients with low socioeconomic status to warrant access to foods and recommended supplements. 

A crucial part of this approach is represented by the change to discuss with the patients about the adequate use of high-quality proteins and supplementations.

Despite the lack of established guidelines on this topic, some valuable indications, experiences and observational studies are available. 

In 2011, an Italian group published a pioneering paper on pregnancy in 12 CKD patients (from stage 2 to 4), with heterogeneous renal diseases. They underwent a low protein (0.6–0.7 g/kg/per day) diet supplemented with alfa ketoacids. The dietary regimen was well tolerated and the patients who decided to carry the baby to term (11/12) delivered at a median gestational age of 33rd week. Only two babies, from the two CKD stage 4 subjects, were SGA. None of the patients required dialysis, while the 7.5 years follow-up showed that all the babies were well, with normal development [32]. Despite some limitations (difficult CKD assessment in pregnancy; no control group; lack of specific nutritional guidelines), this was the first study that pointed out how a controlled low protein diet could be safe and feasible also in CKD pregnant patients.

In 2014, the same group performed a single-arm open interventional study of a low protein diet in pregnant patients with CKD stages 3–5 or severe proteinuria. They compared a treated group (24 pregnancies), who underwent a vegan-vegetarian (milk allowed) low protein diet (0.6–0.8 g/kg per day) plus alfa ketoacids supplements vs. a control group (21 pregnancies, in which diet was not suitable for clinical and/or logistic reasons). The two groups were comparable for age, referral, eGFR, and hypertension, while baseline proteinuria was higher in the intervention group. Even in this case, the experimental dietary regimen was tolerated and more interestingly the incidence of SGA babies was significantly lower in the diet group, while during the long-term follow-up, the children did not present health or socialization problems [33]. A subsequent analysis that compared 36 on diet CKD pregnancies vs. 47 CKD control cases on unrestricted diet in a follow-up of 15 years, confirmed that the incidence of SGA and/or extremely preterm babies (<28th week) was significantly lower in on-diet mothers than in the controls, suggesting again a possible positive effect of a plant-diet in CKD subjects [34]. The same group has more recently published a report focused on the same plant-based diet in three pregnant patients with focal segmental glomerulosclerosis. At the beginning of pregnancy, these three patients showed normal renal function and variable degrees of proteinuria. Then, they followed a moderately restricted plant-based diet with a protein intake of 0.6–0.8 g/kg per day, plus essential amino acids and ketoacids supplements. Remarkably, the adoption of this diet was associated with stabilization or a reduction of proteinuria, without modifications in renal function and serum albumin, while the three babies were born at term, without major issues [35]. An interesting anecdotical experience comes from a Mexican group. A CKD stage 4 31-year-old woman, with a history of miscarriage, underwent dialysis (note that in Mexico access to dialysis is limited by coverage issues) for some months after a preterm delivery and early post-natal death. Renal function recovered slowly to the pre-pregnancy values and then she was started on a moderately restricted low protein, plant-based regimen, employing typical Mexican foods. Subsequently, she became pregnant again, and against medical advice, she decided to not abort and continued a self-managed diet, refusing any further biochemical test. Nevertheless, the pregnancy went well and at 37 weeks she delivered a healthy baby, adequate for gestational age, while her renal function remained stable [36]. 

The authors, admitting the uniqueness of the case report, underlined the importance of compliance and self-empowerment, especially in developing countries, where CKD pregnant patients have to face clinical, economic and logistic challenges, and confirm the safety of the plant-based supplemented diets in pregnancy. Regarding the possible protective mechanisms of low protein diet, especially if supplemented with amino acids and ketoacids, it has been proposed that this kind of diet may contribute to improve uteroplacental circulation and reduce pregnancy-related glomerular hyperfiltration, allowing more babies to reach a later pregnancy stage and reducing the CKD progression rate [35]. In addition, a low protein diet could reduce oxidative stress and protect against endothelial dysfunction, as it has been shown in some experimental models (5/6 nephrectomized rats) [37]. 

## 4. Stage 5 CKD Women Treated with Maintenance Dialysis

### 4.1. Pregnancy in Women on Dialysis: A Still Challenging Condition

Pregnancy in women on maintenance dialysis (HD and PD) is still an uncommon and challenging condition [38], even if registry data indicate that the frequency of this event appears to be increasing. Different reasons may account for the difficulty to conceive, including the reduction of fertility, the presence of sexual dysfunction, and psychosocial factors [39].

In particular, fertility in women with CKD diminishes as the glo-merular filtration rate declines, with the development of menstrual cycle ir-regularities and anovulation, mainly because uremia leads to dysregulation of the hypothalamic-pituitary-gonadal axis. In addition, physical and emotional factors may be responsible for a reduction of sexual desire [40]. In any case, it should be underlined that, although the general incidence of pregnancy is low, it is still lower in women on PD compared with HD (1.06 vs. 2.54 pregnancies per 1000 patient-years) [41]. This may sound strange since theoretically PD provides a more constant biochemical and extracellular environment and it is associated with fewer maternal hypotensive episodes, moreover, patients on PD often have higher residual renal function than HD patients. One of the theories to explain this data is that the hypertonic dialysate in the peritoneum interferes with ovum transit to the uterus [42]. However, as astated above, recent data indicates pregnancy in women on dialysis has become more common than in the past. In particular, data from the Australian and New Zealand Dialysis and Transplant (ANZDATA) Registry confirm that pregnancy rates have increased from the rate of 0.67 pregnancies per 1000 persons in theyears from 1986–1995 to 3.3 pregnancies per 1000 persons in the years 1996–2008 [43]. This increase requires a deeper awareness of the clinical problems specifically related to this particular condition, also considering that, despite improved live birth rates, the babies from dialysis patients are often preterm with a low gestational weight. In this regard, nutritional assessment is crucial to address correct maternal health and fetal growth and development [44].

### 4.2. Principles of Nutritional Management in Pregnant Women on Dialysis

Pre-pregnancy counseling in women on dialysis should warrant the presence of clinical stability with the achievement of blood pressure control and avoiding exposure to teratogenic medications [44]. The energy requirement is similar to that presented by CKD non-dialysis women. A significant exception is constituted by women on PD, in whom, due to calories provided by dialysate glucose absorption, it is recommended a lower energy diet compared with patients on HD (i.e., about 25 Kcal/kg/day plus pregnancy-energy cost) [31].

Beyond global energy requirements, also in dialysis patients, it is important to assess the protein intake. Considering the loss of protein during the dialysis treatment (10–15 g amino acids can be lost daily in the dialysate) [45], that may be further increased by intensive dialysis often used in pregnant women, a high protein intake of about 1.2 g/kg +10 g/day in pregnant HD and PD patients has been suggested [46,47]. The increased calories and protein intake should be accompanied by a progressive increase of the dry weight, considering that a weight gain of 1–1.5 kg during the first trimester and 0.5 kg/week from the second trimester can be expected [11]. Therefore, the fluid balance could be difficult to manage and should be determined individually, taking into account also the residual renal function and the type/frequency of renal replacement therapy provided. Furthermore, supplementation of dialyzable, water-soluble vitamins and minerals is essential in early pregnancy. Indeed, maternal deficiencies in micronutrients can have lasting detrimental effects on the fetus’ organogenesis. Micronutrients to be supplemented include folic acid, vitamin C, thiamine, riboflavin, niacin, vitamin B6 and zinc [47,48].

Folic acid, which is required for neural tube closure, is removed more efficiently by high-intensity hemodialysis, so at least 2 mg of folic acid should be prescribed daily, even if some authors suggest a minimum dose of 5 mg/day. Supplementation of vitamin B12, which low levels have been related to impaired neurodevelopment, could also be useful [49]. On the contrary, supplementation of Vitamin A is not recommended (except in countries where vitamin A deficiency is a public health issue), because, due to the decreased kidney function, there is the possibility of accumulation, which has been related to spontaneous abortion and birth defects [50]. 

Treatment of mineral metabolism disorders, which is often unsuccessful in HD patients [51], comprises the prescription of 1,25(OH) vitamin D to treat secondary hyperparathyroidism and/or for vitamin 1,25 deficiency. However, its use should be guided by levels of parathyroid hormone, calcium, and phosphate, also considering that as a consequence of 1-alpha-hydroxylase placental activity requirements of activated vitamin D may decrease [52]. Paralleling, 25(OH) vitamin D levels should be assessed each trimester and supplemented if low, considering that, if necessary a 25(OH) vitamin D doses of 1000–2000 IU/day is considered safe during pregnancy [48].

Regarding electrolyte supplies, it has been showed that calcium requirement is around 1500–2000 mg/day, while recommended phosphate intake, as gained from recommendations for the healthy pregnancy, is about 1000–1250 mg/day [53]. Mineral homeostasis during pregnancy is important, since it may influence fetal skeletal development. So, for example, maternal hypercalcemia can suppress fetal parathyroid function and cause neonatal hypocalcemia. Therefore, the target of nutritional therapy in pregnant women (with or without dialysis) is to maintain calcium and phosphate levels in the normal range [54]. Daily potassium and sodium intake of 50 mmol and 80 mmol (about 3 and 4 g/day), respectively, are suggested for pregnant women on dialysis [55]. However, the intake of these electrolytes should be individualized according to their serum concentrations, blood pressure, medications, hydration status. Finally, iron supplementation could be required to achieve adequate iron stores, also to prevent anaemia that is related to a higher risk of prematurity and adverse perinatal outcomes. In pregnancy, both oral and iv iron has been safely used [56]. Table 1 summarizes nutrient supplementation required in pregnant women on dialysis.

### 4.3. Clinical Experiences

Once the nutritional needs are established, a nutrition plan, elaborated by a multidisciplinary team, should be implemented taking into account the peculiarities of these patients, including the possible comorbidities (e.g., hypertension, diabetes, etc.) and the risk of adverse fetal outcomes [57]. 

In this regard, the elaboration of a structured medical nutrition therapy by a renal dietician has been proved as effective in the management of pregnant women in dialysis [58]. 

However, beyond dietetic advice, the dialysis treatment per se could also be part of the nutritional management. Interestingly, in a large cohort of pregnant women on dialysis, it has been found that intensive dialysis, when compared with standard dialysis treatment, was correlated with a longer gestational age (36 vs 27 weeks) and an improved live birth rate (85% vs. 48%) [59]. 

Coherently, in a meta-analysis, that included 681 pregnancies in 647 women, Piccoli et al. found a direct relationship between the hours of dialysis per week in HD and lower rates of preterm delivery and SGA [60]. Therefore, some authors suggest that, once the diagnosis of pregnancy has been made, HD time should be increased to at least 20 hours per week, even if others recommend a minimum of 36 h a week in women without residual kidney function, aiming to maintain a predialysis blood urea nitrogen <50 mg/dL, to optimize birth weight and gestational age [61,62].

For this purpose, haemodiafiltration and nocturnal hemodialysis may also be useful [63,64]. Similarly, PD should also be intensified or, alternately switching to HD could be considered [65]. More frequent and intensive dialysis might present several advantages in the management of pregnant women. These advantages include better fluid and blood pressure control, which can be of help in avoiding intradialytic hypotension and achieve the scheduled weight gain [66].

Then, intensive dialysis allows a more liberal diet intake, so that usually dietary restrictions, common in dialysis patients, are not required in pregnant women, who otherwise should often be treated with nutritional supplementations [60]. 

This is also the reason why phosphate binders are often discontinued in these patients. Anyway, in case of hyperphosphatemia, calcium-based binders and vitamin D analogs are considered safe in pregnancy, whereas there are minimal data for other commonly used agents, including sevelamer and lanthanum or the calcimimetics [67]. 

Nevertheless, when intensive dialysis is prescribed, it should be taken into account there is increased protein, vitamin, and electrolytes removal, so nutritional supplementations should be tailored to the individual requirements [68]. Dialysate composition is an additional factor that should be considered in the nutritional management of women on dialysis, especially for its possibility to influence electrolyte levels. So, potassium, phosphate and calcium concentrations in the dialysate can be modified, according to the individual needs [69]. In particular, it has been recommended the use of high calcium bath concentration (1.5 mmol/L) to ensure adequate calcium for fetal skeletal development, especially in the third trimester [70]. 

In Table 2, we report the strategies and suggestions to support nutritional status in pregnant women on dialysis.

## 5. Kidney Transplant

### 5.1. Pregnancy after Renal Transplantation

Kidney transplant (KT) is associated with the rapid restoration of the hypothalamic-pituitary-gonadal axis, leading to improved fertility. So, pregnancy after kidney transplant is more common than in dialysis, even if it is recommended to delay conception for at least 1 year after transplantation, to allow the stabilization of the renal function and drug therapy [71]. 

Moreover, in this case, a high rate of preterm birth and low birth weight has been reported, even if later child development seems appropriate [72,73]. However, clinical experience in this context is limited and some studies are ongoing to better evaluate the incidence and risk factors of adverse pregnancy and transplant-related outcomes [74]. 

Of note, in case of pregnancy in renal transplanted patients, a multidisciplinary pre-pregnancy counseling is necessary to discuss the possible complications, to assess metabolic profile and avoid exposition to teratogenic drugs (for example mycophenolate mofetil) [75].

### 5.2. Nutritional Management in Pregnant Kidney Transplant Patients

Few data have been reported on the specific issue of nutrition in pregnant KT patients. So, general advice can be borrowed by those established for the general population or patients with CKD. However, KT patients present some peculiarities that should be taken into account when considering nutritional prescription [76]. In particular, KT implies the use of immunosuppressive medications, that present known metabolic and nutrition side effects. So, corticosteroids may be associated with an increased risk of gestational diabetes mellitus, increased appetite, weight gain, dyslipidemia, and preterm delivery [77]. Similarly, calcineurin-inhibitors (cyclosporine and tacrolimus), although not teratogenic, have been associated with metabolic alterations, even if with some distinctive features (for example, tacrolimus is more diabetogenic than cyclosporine) [78]. 

Moreover, mTOR inhibitors (i.e., sirolimus, everolimus) may also present metabolic side effects, in particular, hyperlipidemia [79]. Other alterations found in patients taking these drugs include osteopenia, hyperuricemia, hyperkalemia, etc. So, it is clear that a nutritional strategy for pregnant KT patients should be individualized according to the immunosuppressive treatment and biochemical examinations, in particular, monitoring serum glucose and lipid levels [80]. In general, it is recommended an energy supply of about 30 Kcal/kg/die with a protein intake of 1 g/kg/die. In case of reduced renal function, even in RT patients, it is conceivable that a protein-restricted diet, such as advised in CKD patients, should be suitable, even if there are no data [75].

For vitamins and micronutrients, the advice for the general population should be followed.

Instead, for electrolytes, the need for supplementation should be guided by lab examinations. Particular attention should be paid to the supplementation of calcium and 1-25 (OH) vitamin D since KT recipients are at increased risk of developing osteopenia, osteoporosis, and fractures [81,82].

## 6. Conclusions

Appropriate nutritional management is crucial to promote a healthy pregnancy, permit positive fetal outcomes and also influence the offspring’s adult life. However, it remains a challenge in pregnant women with renal diseases, mainly for the need to balance the requirements of the pregnancy with that from the renal dysfunction. So, a better understanding of the biological processes regulating this complex interplay could be of help in addressing specific interventions. The complexity of this condition, also considering the variability of CKD patients and the specific requirements of the different patient population, warrants the necessity of a multifaceted approach, including adequate nutritional and dialysis prescription, social support, and drug management, aiming to improve maternal health and fetal outcomes (see Figure 1).

Eventually, all these kinds of approaches are strictly related and require continuous monitoring and discussion, involving both the health care team and the patients.

## Figures and Tables

**Figure 1 nutrients-12-00873-f001:**
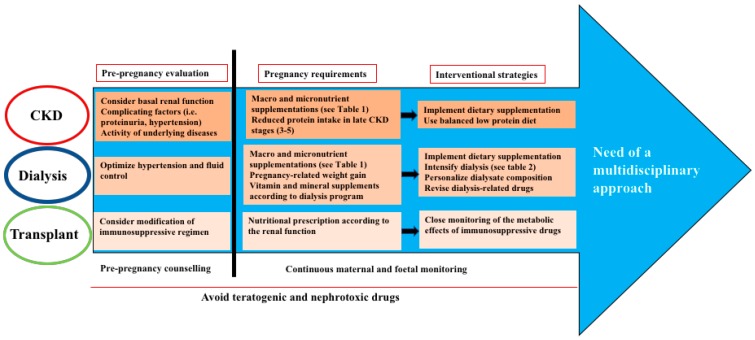
Main clinical issues and recommendations in the nutritional management of pregnant women with chronic kidney disease (CKD), on dialysis and after renal transplant. The multidisciplinary approach should include psychological and social support.

**Table 1 nutrients-12-00873-t001:** Nutritional requirements in pregnant CKD women.

			Non-Dialysis (CKD Stages 3–5)	HD	PD
**Macronutrients**					
	Calories (Kcal/kg/day) *				
	Trimester	First	35	30–35	25–30
		Second-Third	30–35 (+300 Kcal)	30–35 (+300 Kcal)	25–30 (+300 Kcal)
	Proteins (g/kg/day) *		0.6–0.8 (+10 g) *if uremic syndrome is not controlled*, *start dialysis*	1.2 (+10 g)	1.2 (+10 gr)
**Micronutrients**					
	Folic acid (mg/day)		6	2–5	
	25-OH vitamin D (IU/day)		1000–2000	1000–2000	
	Zinc (mg/day)		15	15	
	Iron (mg/day)		20–30	20–30	
	Others §				
**Electrolytes**					
	Calcium (mg/day)		<2000	1500–2000	
	Phosphate (mg/day )		CKD stages 4–5: 800–1000	1200	
	Potassium, mEq/L/day (gr)		According to the serum levels	<75 (3 gr)	

Abbreviations: CKD = chronic kidney disease, HD = hemodialysis, PD = peritoneal dialysis. Notes: For CKD patients in stage 1–2 follow the recommendations for healthy pregnancies. * Calculated on pre-pregnancy weight § Including: vitamin C, thiamine, riboflavin, niacin, vitamin B6 and B12. For reference values and a full list of indications, see Ref 31 and 47.

**Table 2 nutrients-12-00873-t002:** Strategies to support nutritional status in pregnant women on dialysis.

Approaches		Recommendations	Notes	Ref
**Pre-pregnancy counselling**		Consider clinical stability, comorbidities, potential teratogen medications and social conditions		[11]
**Diet modifications**	Nutritional assessment	To perform early Consider nutritional habits, economic conditions Define individual nutritional needs	Consider Medical Nutrition Therapy approach	[58]
	Use of supplements	According to general and disease-specific recommendations (see Table 1)		[12,31]
**Dialysis management**	Dialysis dose	Intensify dialysis: HD: at least >20 h/week HD PD: Increase exchanges (not defined)—consider switch to HD	Maintain predialysis BUN < 50 mg/dL Associated with increased nutrient removal	[62,65]
	Fluid management	To schedule according to the expected weigh gain		[11]
	Dialysate composition	Possibility to individualize potassium, calcium and phosphate concentrations	High calcium content often required	[61,69]
**Dialysis-related drug therapy modification**	Phosphate-binders	Often discontinued, according to phosphate serum levels	If necessary use calcium-based phosphate-binders	[67]
	Vitamin D and iron supplementation	Both oral and iv supplements are considered safe	Frequent monitoring of mineral metabolism and anemia	[43]

Abbreviations: HD = hemodialysis, PD = peritoneal dialysis, BUN = blood urea nitrogen.

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
