# Peer review of "Nutritional Challenges in Pregnant Women with Renal Diseases: Relevance to Fetal Outcomes"

_nutrients, 2020, doi:10.3390/nu12030873_

Round 1
Reviewer 1 Report
The authors are presenting an interesting review article focusing on the chronic kidney diseases (CKD) associated with the pregnant women. This review article will give a clear insight on the nutritional challenges for pregnant women with CKD. However, authors have to take the following points in consideration to make better understating review article.
- Authors have explained different conditions like, with dialysis, without dialysis and kidney transplant. It would be catchy and easily digestible to readers If authors provide a comparative flowchart or a model figure for nutritional requirements and nutritional deficiency in these conditions.
- Authors have provided a brief information about general consideration in pregnant women with renal disease, however, it is not enough to connect the further subheadings provided by authors. Authors must build the concept by giving proper introduction and cite recent research articles.
- Most of the citations by authors are review articles. This implies, authors have written review article basing other review articles. For example, the authors have written “However, global risk assessment of CKD pregnant patients is tricky, because CKD clinical features are highly heterogeneous and many renal diseases are relatively infrequent. In this complex setting, strict and multidisciplinary follow-up is mandatory to provide the best balance between maternal and fetal needs, to identify and manage complications and plan delivery”. In this example, authors have making a statement without any proof by citing any research article, instead they cited review article. There are lot of examples can be found like this.
It would be good if the authors present facts by considering recent ongoing research.
Author Response
As suggested by the reviewer, we fully revised our paper (in particular the first part), adding new references, mostly original research papers, to support our work. Moreover, we added a new figure (figure 1) in which we tried to summarize the main issues and recommendations regarding the nutritional management of pregnant CKD women in different conditions. We hope that the amended version of the paper could be considered more useful and interesting for the reviewer and the readers.
Reviewer 2 Report
The review by Esposito et al. is comprehensive and relevant.
I only have very few comments:
- Line 38: "till to" should be erased.
- Line 110: The recommendation of 35/kcal/kg/day must be extra calories ?? This is however not clear. Please clarify in the text.
- Line 206: It sounds as if ketoacidosis is a good thing the way it is written. As an endocrinologist the term ketoacidosis is associated with a life threatening condition, so would it be possible to rephrase?
- Line 343: that should be changed to than.
Author Response
- Line 38: "till to" should be erased.
We corrected, as suggested.
- Line 110: The recommendation of 35/kcal/kg/day must be extra calories ?? This is however not clear. Please clarify in the text.
Actually, it is well recognized that CKD patients have high energy requirements. So, it is widely accepted and recommended and high energy intake of about 30-35 kCal/kg/day. A similar approach has been suggested for pregnant women with CKD. We added an appropriate reference.
- Line 206: It sounds as if ketoacidosis is a good thing the way it is written. As an endocrinologist the term ketoacidosis is associated with a life-threatening condition, so would it be possible to rephrase?
It was a mistake, thank you for the reviewer’s attention. We corrected the period.
- Line 343: that should be changed to than.
We corrected this mistake, as suggested.
We hope that our new amended version of the manuscript match the reviewer's requirements. In the revised text, changes are marked.
Round 2
Reviewer 1 Report
Authors have addressed all my concerns.
Author Response
N/A